# The Outcome of Childhood Immunoglobulin A Nephropathy with Acute Kidney Injury at the Onset of the Disease—National Study

**DOI:** 10.3390/jcm12206454

**Published:** 2023-10-11

**Authors:** M. Mizerska-Wasiak, E. Płatos, J. Małdyk, M. Miklaszewska, D. Drożdż, A. Firszt-Adamczyk, R. Stankiewicz, B. Bieniaś, P. Sikora, A. Rybi-Szumińska, A. Wasilewska, M. Szczepańska, M. Drożynska-Duklas, A. Żurowska, A. Pukajło-Marczyk, D. Zwolińska, M. Tkaczyk, M. Pańczyk-Tomaszewska

**Affiliations:** 1Department of Pediatrics and Nephrology, Medical University of Warsaw, 02-091 Warsaw, Poland; mpanczyk1@wum.edu.pl; 2Scientific Group in Department of Pediatrics and Nephrology, Medical University of Warsaw, 02-091 Warsaw, Poland; emiliaplatos@gmail.com; 3Department of Pathology, Medical University of Warsaw, 02-091 Warsaw, Poland; jagusiamaldyk@wp.pl; 4Department of Pediatric Nephrology and Hypertension, Jagiellonian University Medical College, 30-663 Krakow, Poland; monika.miklaszewska@uj.edu.pl (M.M.); dadrozdz@cm-uj.krakow.pl (D.D.); 5Department of Pediatrics and Nephrology, Ludwik Rydygier Hospital, 87-100 Toruń, Poland; afirst1@wp.pl (A.F.-A.); rstan1@wp.pl (R.S.); 6Department of Pediatric Nephrology, Medical University of Lublin, 20-090 Lublin, Poland; beata.bienias@umlub.pl (B.B.); przemyslaw.sikora@umlub.pl (P.S.); 7Department of Pediatrics and Nephrology, Medical University of Bialystok, 15-269 Bialystok, Poland; arszuminska@gmail.com (A.R.-S.); annwasil@interia.pl (A.W.); 8Department of Pediatrics, FMS in Zabrze, Silesian Medical University, 40-055 Katowice, Poland; szczep57@poczta.onet.pl; 9Department of Pediatrics, Nephrology and Hypertension, Medical University of Gdańsk, 80-211 Gdańsk, Poland; magdalena.drozynska-duklas@gumed.edu.pl (M.D.-D.); aleksandra.zurowska@gumed.edu.pl (A.Ż.); 10Department of Pediatric Nephrology, Wroclaw Medical University, 50-556 Wroclaw, Poland; pukajlo@o2.pl (A.P.-M.); danuta.zwolinska@umed.wroc.pl (D.Z.); 11Department of Pediatrics, Immunology and Nephrology, Polish Mothers Memorial Hospital Research Institute, 93-338 Lodz, Poland; marcin.tkaczyk45@gmail.com; 12Department of Pediatrics, Immunology and Nephrology, Medical University of Lodz, 92-215 Lodz, Poland

**Keywords:** IgA nephropathy, AKI, children

## Abstract

**Introduction:** IgA nephropathy (IgAN) is the most common glomerulonephritis worldwide. Decreased glomerular filtration rate is a known risk factor for disease progression. **Aim:** We aimed to examine factors that may contribute to disease progression in children that present with impaired eGFR at the onset of IgAN. **Materials and methods**: Of the 175 patients with IgAN from the Polish Registry of Children with IgAN and IgAVN, 54 (31%) patients with IgAN who had an onset of renal function impairment (GFR < 90 mL/min) were eligible for the study. All of them were analyzed for initial symptoms (GFR according to Schwartz formula, creatinine, proteinuria, IgA, C3), renal biopsy result with assessment by Oxford classification, treatment used (R—renoprotection, P—prednisone+R, Aza—azathioprine+P+R, Cyc—cyclophosphamide+P+R, CsA—cyclosporine+P+R, MMF—mycophenolate mofetil+P+R), and distant follow-up. Based on the GFR score obtained at the end, patients were divided into two groups: A—GFR > 90 mL/min and B—GFR < 90 mL/min. **Results:** In the study group, the mean age of onset was 12.87 ± 3.57 years, GFR was 66.1 ± 17.3 mL/min, and proteinuria was 18.1 (0–967) mg/kg/d. Renal biopsy was performed 0.2 (0–7) years after the onset of the disease, and MESTC score averaged 2.57 ± 1.6. Treatment was R only in 39% of children, P+R in 20%, Aza+P+R in 28%, Cyc+P+R in 9%, CsA+P+R in 7%, and MMF+P+R in 3%. The length of the observation period was 2.16 (0.05–11) years. At the follow-up, Group A had 30 patients (56%) and Group B had 24 patients (44%). There were no significant differences in any of the other biochemical parameters (except creatinine) or proteinuria values between the groups and the frequency of the MESTC score ≥ 2 and <2 was not significantly different between Groups A and B. Patients with normal GFR at the follow-up (Group A) were significantly more likely to have received prednisone and/or immunosuppressive treatment than those in Group B (*p* < 0.05) **Conclusions:** In a population of Polish children with IgAN and decreased renal function at the onset of the disease, 56% had normal GFR in remote observation. The use of immunosuppressive/corticosteroids treatment in children with IgAN and impaired glomerular filtration rate at the beginning of the disease may contribute to the normalization of GFR in the outcome, although this requires confirmation in a larger group of pediatric patients.

## 1. Introduction

IgA nephropathy (IgAN), also known as Berger’s disease, is the most common glomerulonephritis worldwide, characterized by the predominance of immunoglobulin A (IgA) deposits in a renal biopsy [1]. The prevalence rate of IgAN in Europe is 2.53/10,000, making the disease a rare one [2].

IgAN might lead to end-stage renal disease (ESRD) in 20–30% of patients after 20 years of disease [3]. In contrast, other studies report that 30% of patients with proteinuria 0.44–0.89 g/g and 20% of patients with proteinuria < 0.44 g/g developed renal failure within 10 years of disease onset [4].

Risk factors for progression of IgA nephropathy include decreased glomerular filtration rate (GFR), sustained hypertension, or substantial proteinuria [5]. Many studies found relevant connections between ESRD and baseline kidney function. They highlighted, i.e., high risk for ESRD among the patients with IgA nephropathy and impaired kidney function at the onset of the disease. Decreased GFR might occur suddenly as acute kidney injury (AKI) and be a signal of insufficiency of the compensation mechanism. It may also emerge at advanced stages of renal damage when damage processes are already irreversible [6]. In the VALIGA study, the multivariate analysis of the 1130 study population, eGFR, MAP, and proteinuria at the time of renal biopsy were significantly associated with the rate of renal function loss [7].

Consequently, not every factor has the same value. While an impaired GFR tends to be a sign of far-progressed kidney damage, proteinuria or hematuria can signal damage in progress. All the above-mentioned factors are closely related to the histopathological lesions found in the renal biopsy, which we rate using the Oxford MEST-C classification [8]. All parameters assessed by this scale are also relevant risk factors.

Worldwide recommendations for the treatment of IgAN in children are not yet established.

The 2021 KDIGO recommendations mainly apply to the adult group, due to the lack of randomized controlled trials in children, which may determine management in the pediatric group [9].

Considering the importance of impaired renal function at the onset of the disease for the prognosis of children with IgA nephropathy, this particular group may be a useful source of knowledge about the natural course of the disease and its treatment.

We aimed to examine factors that may contribute to disease progression in children that present with impaired eGFR at the beginning of IgAN.

## 2. Materials and Methods

Of the 175 patients with IgAN from the Polish Pediatric Registry of IgAN and IgAVN between 2000 and 2020, 54 patients with IgAN were included in the study. They had an onset of disease with impaired renal function (GFR < 90 mL/min), which was considered as acute kidney injury (AKI). All patients had a diagnosis of IgA nephropathy based on renal biopsy with histological confirmation of the predominance of IgA deposits in the mesangium. All biopsy samples were examined by light, electron, and immunofluorescence microscopy. We analyzed the initial parameters and the results of the renal biopsy, with assessment via the Oxford classification, the treatment used, and the follow-up. Based on the GFR score obtained at the follow-up, patients were divided into 2 groups: A—GFR > 90 mL/min and B—GFR < 90 mL/min.

### 2.1. Clinical Features and Biochemical Parameters

Parameters assessed at the start of the disease were age, duration of follow-up, protein in 24-h urine collection, protein and erythrocytes in urinalysis, eGFR, creatinine, protein, IgA, and complement components C3 and C4. After treatment, parameters were measured again.

A level of protein in the urine ≥ 50 mg/kg/day was used as the definition of nephrotic proteinuria and non-nephrotic proteinuria was defined as <50 mg/kg/day. The concentration in the urine sample was measured using the Exton method. Hematuria was determined by more than 5 erythrocytes per high power field in the microscopic urine sediment. Hematuria was indicated by the presence of a change in urine color. The dry chemistry test (Vitro, Ortho Clinical Diagnostic) was used to measure serum creatinine concentration, expressed in mg/dL. GFR (mL/min/1.73 m^2^) was calculated using the Schwartz formula. The concentration of IgA and complement components C3 and C4 were measured by nephelometry at five clinical centers and by turbidimetry at three centers, but the age-dependent normal ranges did not differ significantly between each other. The reason for using two laboratory methods in the assessment of IgA and C3 and C4 concentrations was due to the retrospective nature of the study, which was conducted at several different clinical centers.

### 2.2. Histological Parameters

A renal biopsy was performed to establish the diagnosis. We obtained the diagnostic material from a large-needle percutaneous biopsy performed under ultrasound guidance and evaluated 3 bioptates under the light/electron microscope or in the immunofluorescence tests. We evaluated 25–50 serial slices 2–5 μm thick and performed direct immunofluorescence with fluorescein-conjugated antibodies (FITC). Retrospectively, renal biopsy was assessed using the Oxford MEST-C classification and the overall score was calculated as the sum of M, E, S, T, and C. Assessment criteria: M0 > 50%, M1 < 50%; E—endocapillary hypercellularity: 0—absent, 1—present; S—segmental sclerosis/adhesion: 0—absent, 1—present; T—tubular atrophy/interstitial fibrosis: T0 0–25%, T1 26–50%, T2 > 50%; C—crescents: C0 0%, C1 0–25%, C2 > 25%.

### 2.3. Treatment

Once the disease was diagnosed, patients received various treatments. These were R—renoprotection (ACEI/ARB), P—prednisone+R (renoprotection), Aza—azathioprine+P+R, Cyc—cyclophosphamide+P+R, CsA—cyclosporine+P+R, or MMF—mycophenolate mofetil+P+R. Treatment with Aza, Cyc, CsA, or MMF +P+R was analyzed as I (immunosuppressive treatment).

### 2.4. Follow Up

At the follow-up, blood and urine tests were repeated. Data on the disease progression were submitted anonymously to the Polish Pediatric Registry of IgAN and IgAVN.

Based on the GFR result obtained at the end, patients were divided into 2 groups: A—GFR > 90 mL/min and B—GFR < 90 mL/min.

The study was approved by the Bioethics Committee of the Medical University of Warsaw (No. KB/147/2017).

Figure 1 presents a flow diagram of the study.

## 3. Statistics

Statistical analysis was performed using Dell Statistica 13.0 PL software. Results were presented as mean and standard deviation (SD) for normally distributed variables and as median and range for non-normally distributed variables. The normality of the distribution was checked using the Lilliefors and Shapiro–Wolf test. The statistical significance of differences between mean values was tested using ANOVA for variables with a normal distribution and the Kruskal–Wallis test for variables with a non-normal distribution. The statistical significance of differences between the two groups was calculated using the Student’s *t*-test (for variables with normal distributions) and the Mann–Whitney test (for variables with non-normal distributions). Student’s *t*-test and Wilcoxon test (for normal and non-normal distributions, respectively) were used to test for differences between baseline and follow-up values. A value of *p* < 0.05 was considered statistically significant.

## 4. Results

Patients qualified for the study group represented 31% of the children included in the Polish Pediatric Registry of IgAN and IgAVN (35 boys, 19 girls).

The characteristics of the study group are shown in Table 1.

The mean age of the patient at the onset of IgAN was 12.87 ± 3.57 years, proteinuria was 18 (0–967) mg/kg/d, and mean GFR was 66.12 ± 17.32 mL/min. The follow-up time in the study group was 2.16 (0.05–11) years.

In 72% of children, treatment was P+R or I+P+R (prednisone+renoprotection or immunosuppression+prednisone+renoprotection, respectively).

The prevalence of particular urinary findings is shown in Table 2.

The most common symptom of disease onset was non-nephrotic proteinuria and erythrocyturia, which was observed in 55% of the patients. Gross hematuria was detected in 33% of the patients.

Study parameters were consecutively analyzed in Groups A (GFR > 90 mL/min at the follow-up, 30 pts: 16 boys, 14 girls) and B (GFR < 90 mL/min at the follow-up, 24 pts: 19 boys, 5 girls).

Analysis of clinical signs at the onset showed that the prevalence of nephrotic proteinuria, non-nephrotic proteinuria, hematuria, and hypertension were not significantly different between groups, as shown in Table 2.

There were neither significant differences in any of the other biochemical parameters studied nor differences in proteinuria levels between the groups. Median proteinuria and mean creatinine levels were higher, and GFR was lower in Group A, although not significantly, as shown in Table 3.

We analyzed the Oxford kidney biopsy classification findings in Groups A and B. There were no significant differences in the mean MESTCscore in Groups A and B, but a MESTCscore > 3 was present in 33% of Group B patients and 17% of children in Group A, as shown in Figure 2.

We also reviewed the methods of treatment used in Groups A and B in the evaluation of long-time follow-up.

Patients with normal GFR at the follow-up (Group A) were significantly more likely to receive prednisone with renoprotection and/or immunosuppressive treatment with renoprotection than those in Group B (*p* < 0.05), as shown in Table 4.

## 5. Discussion

Our retrospective study shows the outcome of pediatric patients with IgAN whose disease started with acute kidney injury. Such a group of children with IgAN was not previously described in the medical literature. This may relate to a situation of acute onset, but also to an incidentally recognized abnormal GFR combined with urinary alterations. This is a result of the natural course of the disease, which can remain undiagnosed for several years.

According to Japanese evidence and the authors’ own research, the average age of diagnosis of the disease in pediatric population is around 11 years, although in Chinese studies diagnosis is even below 16 years of age [10,11,12]. A greater age of diagnosis may be a risk factor for poor prognosis. In our previous studies, such an age was >13.9 years [11]. In the current study, the age of diagnosis in Group B patients (GFR < 90 mL/min at follow-up) was also above 13 years.

In a Polish study group, AKI was found in 31% of children with IgAN, while in the Chinese cohort, AKI with hematuria and massive proteinuria was detected in 9.7% of 196 children with IgAN [13], C1 was found in 80%, E1 in 60%, and T1-2 in 70% [12], whereas in our group C1-2 was 37%, E1 39%, and T1-2 in 30% of patients.

Proteinuria at the onset of the disease was a median of 15 mg/kg/day (0–220) and was not significantly different from that in Group A of 28 mg/kg/day (0–967). At the end of follow-up, a reduction in proteinuria was obtained in both groups (with no significant statistical difference in median values), but the GFR in Group B remained abnormal after a follow up period of 2.16 (0.05–11) years.

According to the findings of Pitchner et al., even a small persistent proteinuria < 0.44 g/g may impair renal function at distant follow-up [4]. Nevertheless, the authors of this article looked for factors differentiating these two groups of patients with an unfavorable prognosis, due to the presence of reduced GFR at the onset of the disease as an independent factor of poor prognosis.

Among the analyzed treatment methods, the use of prednisone in combination with renoprotection or immunosuppression in 72% of patients in the study group may be an indicator of disease severity.

Noteworthy is the fact that patients in Group A (with normal renal function at the follow-up) had, at the start of the disease, higher proteinuria, higher creatinine, and lower GFR, and although these differences were not statistically significant, they were marked in relatively small groups of patients.

According to Coppo et al., in children with IgAN, contrary to adults, it is not common to find slowly progressive cases of IgAN that can only be treated with renoprotective therapy for 6 months, as recommended by KDIGO. Pediatric nephrologists are aware of the possibility of progression that is not completely blocked by renin-angiotensin system blocking (RASB) drugs and require more aggressive anti-inflammatory treatment with CS/IS (corticosteroids/immunosuppressive) drugs [3].

In the VALIGA Study in 261 children and young adults, the use of corticosteroid and/or immunosuppressive therapy (CS/IS) was confirmed in more than 50% of children. The study also showed that children < 16 years old with M0 and normal renal function at the onset of disease had a high probability of resolution of proteinuria at the end of treatment and the benefits of CS/IS therapy were statistically significance [14].

In our work, we showed that a significant factor distinguishing the group with a good outcome from the group with a poor outcome was the use of prednisone in combination with renoprotection or immunosuppression which occurred significantly more often in Group A. This could confirm the thesis that immunosuppressive treatment/steroid therapy should be started earlier in children than in adults, especially in the group of children with impaired renal function at the beginning of the disease, with the presence of non-nephrotic proteinuria.

Similar conclusions were reached by Japanese authors who, in a randomized control trial, demonstrated the efficacy of two years of CS/IS therapy in children with IgAN combined with antiplatelet treatment [15].

Cabier et al. confirmed positive results of corticosteroids and/or immunosuppressive treatment in an uncontrolled study group of children from France of 6 months duration. This group included patients with acute kidney injury, macroscopic hematuria, or acute nephritic syndrome [16].

The use of CS/IS treatment in the pediatric population is still a subject of discussion, but it seems that it might be beneficial for this group.

## 6. Limitation

The main limitation of the study is the relatively small study group. Nevertheless, considering the rarity of the disease, obtaining data from a national registry, and inclusion of the patients with AKI only, it still represents a large group. The results need to be confirmed in a larger, international group of children. Other limitations of the study are the retrospective design and lack of a control group. A prospective group-controlled study with predefined treatment and follow-up criteria would be needed to provide stronger conclusions.

Our research does not evaluate treatment methods, but in the absence of treatment standards in children, these data were considered important in the analysis of outcome in children. It is also important to mention that only patients with renal dysfunction were included, which is not representative of all pediatric patients with IgA nephropathy. We are also aware that, in addition to the treatment we mentioned, there may have been other disturbing factors and effects on renal function.

## 7. Conclusions

In the Polish pediatric population with IgAN and impaired renal function at the onset of the disease, a normal GFR in the follow-up is observed in 56% of the cases.

The use of immunosuppressive/corticosteroid treatment in this group may contribute to the normalization of GFR in long-term follow-up, which requires confirmation in a larger group of pediatric patients.

## Figures and Tables

**Figure 1 jcm-12-06454-f001:**
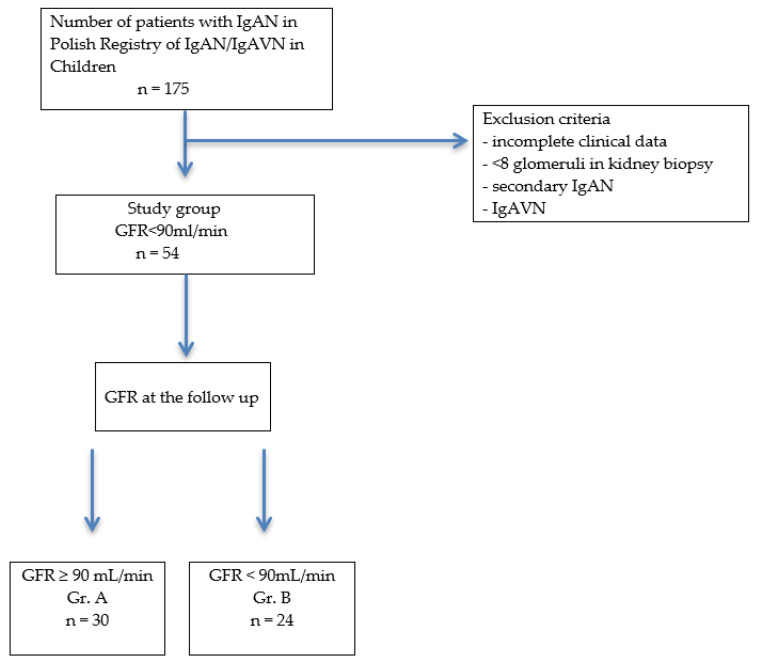
Flow diagram of the study.

**Figure 2 jcm-12-06454-f002:**
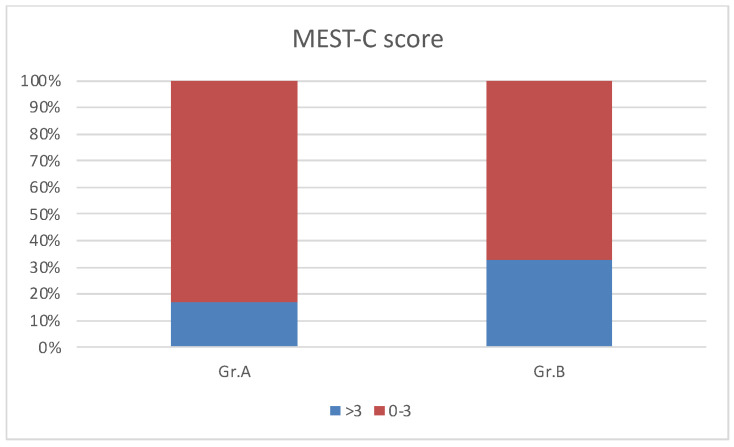
Results of kidney biopsy in Oxford classification in Groups A and B.

**Table 1 jcm-12-06454-t001:** Characteristics of study group.

Parameters	
Age at the onset (years)	12.87 ± 3.57
Proteinuria at the onset啵(mg/kg/day)	18 (0–967)
Creatinine at the onset (mg/dL)	1.03 ± 0.4
GFR at the onset (mL/min)	66.12 ± 17.32
IgA at the onset (mg/dL)	296.3 ± 131.95
C3 at the onset (mg/dL)	117.41 ± 21.87
C4 at the onset (mg/dL)	24.97 ± 8.59
Age at the follow-up (years)	17.41 (8.85–19.35)
Proteinuria at the follow-up (mg/kg/day)	0 (0–370)
Creatinine at the follow-up (mg/dL)	0.77 ± 0.24
GFR at the follow-up (mL/min)	93.15 ± 21.96
IgA at the follow-up (mg/dL)	266.03 ± 133.28
C3 at the follow-up (mg/dL)	110.06 ± 20.84
C4 at the follow-up (mg/dL)	25.21 ± 19.96
Treatment without medication R only P+R I (Aza/Cyc/Csa/MMF+P+R) Aza+ P+R Cyc+ P+R CsA+ P+R MMF+ P+R	4 (7%) 20 (39%) 11 (20%) 26 (52%) 15 (28%) 5 (9%) 4 (7%) 2 (3%)

R—renal protection; P—prednisone; I—immunosuppressive therapy; Aza—azathioprine; Cyc—cyclophosphamide; CsA—cyclosporine; MMF—mycophenolate mofetil; GFR—glomerular filtration rate.

**Table 2 jcm-12-06454-t002:** Clinical symptoms at the onset of the disease in Groups A and B.

	N (%)	A N = 30	B N = 24	
Nephrotic proteinuria	15 (28%)	8	7	NS
Non-nephrotic proteinuria	30 (55%)	18	12	NS
Gross hematuria	18 (33%)	8	10	NS
Hypertension	17 (31%)	8	9	NS

**Table 3 jcm-12-06454-t003:** Comparison of parameters at the onset and at the follow up between Groups A and B.

	A (*n* = 30) GFR ≥ 90	B (*n* = 24) GFR < 90	*p*
Age at the onset (years)	12.69 ± 4.2	13.12 ± 2.93	NS
Proteinuria at the onset (mg/kg/day)	28 (0–967)	15 (0–202)	NS
Creatinine at the onset (mg/dL)	1.09 ± 0.5	0.95 ± 0.23	NS
GFR at the onset (mL/min)	62.95 ± 19.38	71.72 ± 12.77	NS
IgA at the onset (mg/dL)	252.74 ± 92.18	334.81 ± 120.35	NS
C3 at the onset (mg/dL)	112.43 ± 18.47	119.36 ± 19.71	NS
C4 at the onset (mg/dL)	25.99 ± 7.8	22.52 ± 8.29	NS
Age at the follow-up (years)	15.55 (8.85–19.35)	17.48 (9.78–18.92)	NS
Proteinuria at the follow-up (mg/kg/day)	4.15 (0–370)	0 (0–73.3)	NS
Creatinine at the follow-up (mg/dL)	0.64 ± 0.13	0.95 ± 0.25	NS
GFR at the follow-up (mL/min)	108.62 ± 16.19	73.92 ± 11.09	NS
IgA at the follow-up (mg/dL)	222.7 ± 112	304.65 ± 123.45	NS
C3 at the follow-up (mg/dL)	109.28 ± 19.94	113.61 ± 23.08	NS
C4 at the follow-up (mg/dL)	27.58 ± 26.72	22.06 ± 7.54	NS

R—renal protection; P—prednisone; I—immunosuppressive therapy; Aza—azathioprine; Ex—dexamethasone; Csa—cyclosporine; MMF—mycophenolate mofetil; GFR—glomerular filtration rate; NS—not significant.

**Table 4 jcm-12-06454-t004:** Comparison of the methods of treatment in children with IgAN in Group A and B.

Treatment	Gr.A (GRF≥ 90 mL/min)	Gr.B (GFR < 90 mL/min)	*p*
Without medication	2 (6.7%)	2 (8.3%)	NS
R only	8 (26.7%)	12 (50%)	NS
P+R	8 (26.7%)	3 (12.5%)	NS
P+R or I+P+R	26 (86.6%)	15 (62.5%)	<0.05
I (Aza/Cyc/CsA/MMF+P+R)	16 (53.3%)	10 (41.7%)	NS
Aza+P+R	10 (33.3%)	5 (20.8%)	NS
Cyc+P+R	5 (16.7%)	0	<0.05
CsA+P+R	1 (3.3%)	3 (12.5%)	NS
MMF+P+R	0	2 (8.3%)	<0.05

R—renal protection; P—prednisone; I—immunosuppressive therapy; Aza—azathioprine; Cyc—cyclophosphamide-; CsA—cyclosporine; MMF—mycophenolate mofetil; GFR—glomerular filtration rate; NS—not significant.

## Data Availability

Not applicable.

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
