# Peer review of "The Outcome of Childhood Immunoglobulin A Nephropathy with Acute Kidney Injury at the Onset of the Disease—National Study"

_jcm, 2023, doi:10.3390/jcm12206454_

Round 1

Reviewer 1 Report

The topic is very interesting, and indeed the sample is relatively small, but the authors underlined this in the limitations section. Nevertheless, my main concern regards the comparison between the groups since it does not seem correct to compare the groups with different treatments. In order to obtain valid the treatment should be the same or at least similar in the two groups. Therefore, taking into account the small number of patients in each group, it is impossible to compare the effect of each treatment and to obtain relevant conclusions.

The language requires only editing corrections.

Author Response

Dear reviewer,

we sincerely thank you for your feedback. Indeed, the sample group is small, but IgAN is a relatively rare disease and thus it is difficult to collect a large sample group even in the National Registry. Due to the lack of precise guidelines for the treatment of IgAN in children and the participation of many research centres in the study, the treatment of patients in our study is not standardised but is based on the principals of renoprotection and immunosuppressive treatment to prevent further progression of the disease and to reverse the current inflammatory changes. The type of treatment was only mentioned and proved to be the only significant difference in the study groups. We hope that we have thus cleared up your doubts.

Reviewer 2 Report

Manuscript The Outcome of childhood IgA nephrophaty with acute kidney injury at the onset of the disease - national study represents interesting study with promising results. In the introduction, it is necessary for the authors to expand it and introduce the reader in more detail to the researched issue. Also, it is necessary to add references in the introduction in the parts where they are missing. It is necessary to make better quality figures (Figure 1 and Figure 2) and to expand the discussion and discuss the obtained results in an adequate and more comprehensive way.

Author Response

Dear reviewer,

we sincerely thank you for your comments. We have improved quality of Figure 1 and 2. We have also added the latest relevant literature to the introduction and discussion and expanded the above-mentioned sections of our article. We hope that we have thus cleared up your doubts.

Reviewer 3 Report

In this manuscript, M. Mizerska-Wasiak and colleagues assessed the regression of acute kidney injury at the onset of IgA nephropathy in children. This study, although seemingly interesting, has limitations: the study group was relatively small, and the results of the present study would need to be confirmed in a much larger international cohort of children. The results of this study are not generalizable.

Author Response

Dear reviewer,

we sincerely thank you for your feedback. We strongly agree with your opinion that the study group is small. However, the group only includes patients from the national registry, making it a restricted number of children. We also plan to undertake a comparison of treatment outcomes in children in other countries, and by publishing our study we would like to encourage other researchers to publish their observations of treatment and outcome of IgAN in children. We hope that we have thus cleared up your doubts.

Round 2

Reviewer 2 Report

The authors successfully responded to all objections. 

Author Response

Dear Reviewer, 

thank you very much.